# The Brain–Gut–Microbiome Axis in Psychiatry

**DOI:** 10.3390/ijms21197122

**Published:** 2020-09-27

**Authors:** Seung-Ho Jang, Young Sup Woo, Sang-Yeol Lee, Won-Myong Bahk

**Affiliations:** 1Department of Psychiatry, School of Medicine, Wonkwang University, Iksan 54538, Korea; psychicjang@gmail.com (S.-H.J.); psysangyeol@hanmail.net (S.-Y.L.); 2Department of Psychiatry, College of Medicine, The Catholic University of Korea, Seoul 07345, Korea; youngwoo@catholic.ac.kr

**Keywords:** brain–gut axis, microbiome, psychiatry, neurotransmitters

## Abstract

Beginning with the concept of the brain–gut axis, the importance of the interaction between the brain and the gastrointestinal tract has been extended to the microbiome with increasing clinical applications. With the recent development of various techniques for microbiome analysis, the number of relevant preclinical and clinical studies on animals and human subjects has rapidly increased. Various psychotic symptoms affect the intestinal microbiome through the hypothalamus–pituitary–adrenal gland axis. Conversely, the intestinal microbiome regulates the gastrointestinal tract environment and affects psychological factors by means of the microorganisms or their metabolites, either acting directly on the brain or through the synthesis of various neurotransmitters. This review discusses the clinical applicability of the brain–gut–microbiome axis and directions for improving psychological symptoms based on the studies published to date.

## 1. Brain–Gut Axis

The connection between the brain and gut is known to affect human emotion, motivation, and higher cognitive function, in addition to maintaining homeostasis of the gastrointestinal tract. This complex interaction is referred to as the brain–gut axis (BGA). It coordinates and integrates gastrointestinal function in addition to connecting the brain to peripheral functions [1]. During these processes, mechanisms such as immune response activation, preservation of intestinal permeability, various intestinal responses, and gastrointestinal tract–endocrine signaling occur [2]. Therefore, it can be inferred that signaling between the brain and gastrointestinal tract is based on regulating the neuro-immuno-endocrine system. Such bidirectional signaling systems include the central nervous system (CNS), autonomic nervous system, gastrointestinal nervous system, and hypothalamic–pituitary–adrenal (HPA) axis. The autonomic nervous system, which includes sympathetic and parasympathetic nerves, starts in the lumen of the intestinal canal and reaches the CNS through the enteric, spinal, and vagal pathways. It sends signals from the brain to the intestinal canal and vice versa [3].

The HPA axis is the most critical system for coordinating adaptive responses to various stressors. As a part of the limbic system, it is essential for regulating brain functions, including memory and emotional responses [4]. When the HPA axis is activated due to environmental stress or an increase in proinflammatory cytokines, corticotropin-releasing factor (CRF) is secreted from the hypothalamus, stimulating the secretion of adrenocorticotrophic hormone (ACTH) from the pituitary gland. As a result, cortisol—one of the major stress hormones that affects various parts of the human body including the brain—is secreted from the adrenal cortex. Therefore, the brain can coordinate the function of the gastrointestinal tract via the integration of signaling through nerves and hormones.

## 2. Microbiomes

The microbiome is an ecological community comprised of microorganisms such as bacteria, archaea, microbial eukaryotes, fungi, and viruses living in the human body, as well as their genes [5]. The adult human gut contains approximately 1 kg of bacteria, which is similar to the weight of the brain. In general, 10^13^ to 10^14^ microorganisms are present in the human intestine, which is greater than the number of all human cells combined [6]. They contain 100 times the genes in the human genome. The intestinal microbiome is distributed across the human intestine, and individual differences exist in intestinal microbial profiles, but the relative amount and distribution of bacterial phylotypes in the intestinal tract are similar [7]. To date, *Firmicutes* spp. and *Bacteroides* spp. are known to be the most common, accounting for 3/4 of all intestinal microbiomes [8]. Numerous studies have reported that the distribution of intestinal microorganisms is essential for organs that are responsible for normal physiological functions and intestinal health, especially the brain. From newborns to the elderly, dysbiosis of the intestinal microbiome has profound effects on brain function. Dysbiosis is influenced by childbirth, diet, and drug exposure [9].

In previous decades, human microbiome research faced difficulties due to technical limitations [10]. Following the recent development of microbiome sequencing techniques, it has been proposed that various microorganisms inhabit the human body in large numbers, and this has great relevance to human health and disease [11]. Microbiomes have begun to emerge as highly efficient biomarkers for assessing human health. In particular, they are essential for maintaining normal host physiology, including the formation and development of the immune system [12]. As colonization of the intestinal microbiome is critical for immune system development, an imbalance or deficiency in the intestinal ecosystem changes the immune response, thereby contributing to systemic physiological dysfunction as well as increasing inflammation and oxidative stress, as commonly observed in mental illness [13]. Therefore, the study of microbiomes is essential for understanding the pathogenesis of chronic psychiatric diseases and developing the necessary treatments.

Many studies have reported that gut microbiomes play an important role in the BGA, with direct effects on the CNS through neuroendocrine and metabolic pathways, as well as on gastrointestinal tract cells or the enteric nervous system (ENS). The theory that intestinal microbes affect the CNS in humans originates from a study that reported improved consciousness in patients with hepatic encephalitis following antibiotic treatment [14]. To date, most relevant studies have focused on germ-free (GF) animal, probiotic, antibiotic, and infection studies [15]. In GF animal studies, it was reported that bacterial colonization in the intestine plays a central role in the development and maturation of the ENS and CNS. Compromised colonization of intestinal microbes affects the expression and transformation of neurotransmitters in both the ENS and CNS [16]. Studies in GF animals have shown that gut microbes have the ability to modulate stress responsiveness, anxiety-like behavior, and HPA activity. Further, colonization of microorganisms in the gastrointestinal tract leads to maturation of the HPA axis with age. A study reporting that only the young age group recovered from an excessive stress response after GF colonization demonstrated that there was a critical period for the recovery of neuroplasticity following the influx of intestinal microorganisms [17]. In addition, deficiency in cognitive function was found in GF animals, which seemed to be related to the deficiency of brain-derived neurotrophic factor (BDNF), the most important factor in memory [18].

## 3. Frontiers in Microbiomes

Researchers have focused on how the intestinal microbiome affects the brain. In relation to psychiatric disorders, intestinal microbes are attracting a lot of attention due to their essential role in regulating brain function and behavior through the gut–brain axis [19]. It is not yet known how peripheral gut microbes are related to emotions and cognitive function, but it is hypothesized that interactions occur via multiple routes including gut hormone signaling, the vagus nerve, immune system, microbial metabolites (e.g., short-chain fatty acids), and tryptophan metabolism [20]. Recent studies have used GF animals to directly assess behavioral aspects similar to those of psychiatric disorders, such as depression or anxiety. A study by Sudo et al. revealed, for the first time, that intestinal microbes can coordinate the HPA axis. When GF mice were exposed to mild stress, the secretion of adrenocorticotrophic hormone and corticosterone was increased to a greater extent than in those with a normal intestinal microbiome. This stress response was completely restored via colonization with *Bifidobacterium infantis* and partially recovered when colonization was performed with feces from normal mice [17]. In a study by Clarke et al., anxiety-like behavior decreased in GF mice, and the concentrations of neurotransmitters and neurotrophic factors in the brain changed [21]. Zheng et al. described how the intestinal microbiome affects psychobehavioral characteristics and how a change in the intestinal microbiome can cause depression-like behaviors. In GF mice that were given microflora from the feces of patients with depression, depression-like behavior increased compared with the group that received the feces of healthy subjects [22]. In a study by Kelly et al., GF rats that underwent fecal microbiota transplantation (FMT) with feces from patients that exhibited behaviors and physiological characteristics of depression, including anhedonia, anxiety-like behaviors, and changes in tryptophan metabolism. Therefore, intestinal microbes influence psychological conditions such as depression and anxiety-like behavior through various physiological mechanisms [23].

## 4. Neurotransmitters in Microbiomes

Metabolites produced by gut microbiota can affect the brain–gut axis. Butyrate, a well-known short-chain fatty acid (SCFA), is synthesized by microbiota and inhibits histone deacetylases to promote memory and neural plasticity [24]. In particular, propionate produced by microbiota protects the blood–brain barrier (BBB) from oxidative stress [25]. SCFAs influence neuroinflammation by coordinating the proliferation and recruitment of immune cells, such as T-cells and neutrophils, as well as inflammatory cytokine production.

In addition to SCFAs, various microbiota-derived metabolites act as essential neuroactive molecules in the central nervous system (CNS). *Lactobacillus* spp. and *Bifidobacterium* spp. synthesize neurotransmitters, such as acetylcholine and gamma-amino butyrate (GABA), while *Streptococcus* spp., *Enterococcus* spp., and *Escherichia* spp. synthesize serotonin, dopamine, and norepinephrine [20,26]. However, it has not been clearly identified how neurotransmitters synthesized by microbiota work within the host. Some essential vitamins—such as vitamins K, B2, B9, and B12, synthesized by microbiota—have a neuroprotective effect on the CNS [27]. Previous studies have reported how microbiota regulate the bioavailability of various substances necessary for neurotransmitter synthesis [28,29]. Gut microbes can metabolize the essential amino acid tryptophan as a precursor for the synthesis of indole, serotonin, and melatonin, thereby limiting the availability of tryptophan for the host [30]. *Pseudomonas* spp. synthesizes serotonin from tryptophan and is used for toxicity and intercellular signaling. Therefore, the gut microbiota-associated decrease in circulating tryptophan affects serotonergic transmission and function of the CNS and enteric nervous system, and tryptophan metabolism induced by microbiota affects tryptamine production through the action of tryptophan decarboxylase [31]. Tryptamine modulates the inhibitory response of cells to serotonin by enhancing serotonin secretion from enterochromaffin cells [32]. Gut microbiota-associated tryptophan metabolism yields a number of substances affecting the brain and behavior including kynurenine, quinolinate, indole, and indole derivatives [33]. Kynurenine and quinolinate can affect brain function, leading to depressive symptoms and host tryptophan deficiency [34]. The production of kynurenine and quinolinate reduces tryptophan concentration in the blood and inhibits the production of major neurotransmitters, such as serotonin, in the brain [35]. In addition, indole and indole derivatives, including indole acetic acid (IAA) and indole propionic acid (IPA), alter CNS metabolism [36]. Therefore, tryptophan catabolism associated with microbiota is the most important regulator of the brain–gut axis.

Dopamine is the most important neurotransmitter that coordinates reward-based motivation and plays a precursor role in synthesizing catecholamines, such as norepinephrine and epinephrine. To date, the functions of norepinephrine include arousal and alertness in sensory signal detection and the waking state. In recent studies, it was reported to play an important role in cognitive functions, such as behavior, memory, learning, and concentration [37]. Various microbes are involved in catecholamine synthesis and function. The growth rate of pathogenic *Escherichia coli* 0157:H7 (EHEC) increased in the presence of dopamine and, in particular, norepinephrine, as enhanced motility, biofilm formation, and virulence were observed [38]. The growth of *Klebsiella pneumoniae*, *Pseudomonas aeruginosa*, *Enterobacter cloacae*, *Shigella sonnei*, and *Staphylococcus aureus* also increased in the presence of norepinephrine [39]. *E. coli*, *Proteus vulgaris*, *Serratia marcescens*, *Bacillus subtilis*, and *Bacillus mycoides* acted as harbors of relatively high levels of norepinephrine with regard to biomass [40]. The role of the microbiota in coordinating norepinephrine and dopamine levels in vivo has not yet been confirmed, but it appears to be involved in their biosynthesis/catabolism within the host. In a recent study, the concentration of norepinephrine in the cecal lumen and tissue of GF animals was significantly lower, and the cecal level of norepinephrine recovered when a mixture of microbiota and 46 clostridia species was colonized [41]. These studies suggested that microbiota affect norepinephrine in the lumen, but it is unclear whether the bacteria directly produce norepinephrine or coordinate its production by the host. The effect of microbiota on the catecholamine system is functionally important, and the behavioral response to cocaine increased in microbiota-deficient mice, which may be associated with the increased activity of the D1 dopamine receptor Drd1 and GluR2 AMPA receptor Gria2 [42]. The behavioral response to cocaine normalized when animals that received antibiotics were also supplemented with SCFA or microbial fermentation byproducts, which suggests that the microbiota has an indirect effect on reward behavior.

Gamma-aminobutyric acid (GABA) is an important inhibitory neurotransmitter in the CNS, and its receptors are widely distributed. GABAergic neurotransmission is important for various CNS functions, such as behavior, pain, and sleep, as well as gastrointestinal tract functions, such as intestinal motility, gastric emptying, nociception, and acid secretion [43]. Microbes consume or produce GABA. In terms of consumption, the GABA shunt, which converts GABA to succinate through the TCA cycle, is important. *E. coli* produces GABA from sole carbon and nitrogen sources, but the general function of GABA-consuming microbiota remains unknown. In contrast, the systemic function of GABA has been well described, and various types of microbes are known to synthesize GABA. Unlike other neurotransmitters, GABA production has been studied extensively in terms of physiology, and GABA secretion plays a role in reducing the intracellular pH of the glutamate acid resistance system [44]. The microbiota affects circulating GABA levels, and significant decreases in luminal serum levels have been reported in GF animals [45]. Members of the *Bifidobacterium* and *Lactobacillus* genera are involved in GABA synthesis, and when *Lactobacillus rhamnosus* (JB-1), the most frequently reported one, was injected into mice, depressive symptoms and anxiety decreased in association with the activity of the vagus nerve, and changes in cerebral GABAergic activity were observed [46]. In a recent study, oral administration of *Bifidobacterium breve* NCIMB8807 pESHgadB in a rat model reduced GABA synthesis and sensitivity to visceral pain through the overexpression of glutamate decarboxylase B [47]. There are only a few, similar preliminary studies on this in humans, but tuning the human microbiota is expected to affect GABA levels. Dietary regulation changes the function and composition of the microbiota, and the ketogenic diet is associated with increased levels of GABA in cerebrospinal fluid and symptom improvement in children with treatment-resistant epilepsy [48]. In a recent fecal transplant study, GABA was the most variable metabolite in obese patients who received fecal transplants from lean donors, and this was also associated with improved insulin sensitivity [49].

## 5. Microbiome Research in Psychiatric Disorders

### 5.1. Depression

Various clinical or preclinical studies have been conducted to evaluate the effect of the microbiome on depression. Gareau et al. separated four-to-nine-day-old pups from their mother for approximately three hours every day and compared them with control pups. During the separation period, probiotic organisms (two *Lactobacillus* spp.) were administered. Blood corticosterone concentration was evaluated after 20 days, and the group that received probiotics exhibited a decrease in corticosterone levels [50]. In a study by Bharwani et al., male C57BL/6 mice were given JB-1 for 28 days before assessing chronic social defeat and changes in behavior and immune cell phenotypes. In the group administered JB-1, stress-induced anxiety-like behavior was significantly reduced, and the lack of social interaction was prevented [51]. Burokas et al. reported that continuous FOS + GOS treatment was found to have antidepressant and anxiolytic effects in C57BL/6J male mice receiving prebiotics (fructo-oligosaccharides (FOS) and galacto-oligosaccharides (GOS) [52]. In a study by Desbonnet et al., the motivation state was assessed using the forced swim test (FST) after the administration of bifidobacteria or citalopram in a maternal separation model. In the probiotics group, immune response, behavioral deficits, and basal noradrenaline concentration in the brainstem were all normalized [53]. Bravo et al. reported an increase in GABA B1b mRNA in the cortical regions of the brain along with a decrease in the hippocampus, amygdala, and locus coeruleus following the continued administration of JB-1. In addition, the JB-1 group exhibited a reduction in stress-induced corticosterone as well as anxiety- and depression-related behavior [46]. In a study by Ait-Belgnaoui et al., *L. farciminis* was administered for two weeks in a partial restraint stress (PRS) experiment, and the PRS group had higher plasma ACTH, corticosterone, hypothalamic CRF, proinflammatory cytokine levels, and portal blood concentration of LPS. *L. farciminis* and ML-7 groups induced inhibition of stress-induced hyperpermeability and endoxemia along with the prevention of neuroinflammation and stress response through the HPA axis [54]. In a study by Liang et al., restraint stress was applied to adult specific-pathogen-free (SPF) rats for 21 days before administering *Lactobacillus helveticus* NS8 or citalopram followed by behavioral tests (sucrose preference test, elevated plus maze test, open-field test, object recognition test, and object placement test). The results indicated that anxiety and depressive behavior as well as cognitive dysfunction, were improved in the *L. helveticus* NS8 group at a level similar to that of the citalopram group [55]. In a study by Bharwani et al., the intestinal microbiome was assessed after chronic social defeat in male C57BL/6 mice, and the degree of social deficit was correlated with the microbial community profile. In addition, in silico analysis of the functional profile of microbiomes in the chronic social defeat group revealed that functional diversity was reduced and pathways, including those for the synthesis and metabolism of neurotransmitter precursors and SCFAs, were also downregulated [56]. Kato-Kataoka et al. analyzed psychological characteristics, saliva cortisol, fecal serotonin, and plasma L-tryptophan following administration of *Lactobacillus casei* strain Shirota (LcS). It was also reported that fermented milk containing LcS was useful for improving physical symptoms in stressful situations, based on the significant increase in fecal serotonin levels in the LcS group after two weeks and a significant decrease in physical symptoms after five weeks [57]. Table 1 summarizes the main characteristics of these trials.

### 5.2. Attention-Deficit/Hyperactivity Disorder (ADHD)

ADHD is a common neuropsychiatric disorder with inattention and/or impulsivity and hyperactivity as its main symptoms. Abnormalities in the monoamine neurotransmitter systems of dopamine and noradrenaline are recognized as the main cause of ADHD [58]. Precursors of monoamines involved in ADHD (i.e., dopamine, noradrenaline, serotonin) are produced by several members of the gut microbiota.

Howard et al. retrospectively compared a Western diet with a healthy diet in 115 adolescents who were diagnosed with ADHD. In the group with a Western dietary pattern, there was a significantly higher number of ADHD diagnoses (OR = 2.21, 95% CI 1.18–4.13) [59]. When Aarts et al. conducted 16S RNA gene sequencing in 19 ADHD patients and 77 controls, a difference in the relative abundance of several bacterial taxa was observed, especially in *Bifidobacterium*. In addition, there was an increase in cyclohexadienyl dehydratase in the experimental group compared to the control group, and this enzyme is associated with dopamine precursor synthesis [60]. Partty et al. divided 75 subjects into a group receiving *L. rhamnosus* GG and a placebo group and compared their gut microbiota after 3 weeks, 3 months, 6 months, 18 months, 24 months, and 13 years. After 13 years, ADHD symptoms appeared in 6/35 (17.1%) subjects from the placebo group but in none of the subjects from the probiotics group. Therefore, it was suggested that taking probiotic supplements early in life may reduce the incidence of neuropsychiatric disorders [61]. Table 2 summarizes the main characteristics of these trials.

### 5.3. Autism Spectrum Disorder (ASD)

Autism is a neurodevelopmental disorder with a clearly increasing prevalence. The disorder is characterized by deficiencies in language acquisition and sociability [62]. Autism is often accompanied by digestive symptoms, and this association has long been controversial. Since more than 70% of patients complain of digestive symptoms, the disease is sometimes viewed from the perspective of the brain–gut axis. Borre et al. assessed the behaviors of mice that grew up in a germ-free environment. Mice were assessed in three separate rooms, i.e., a GF mouse was placed in the middle with a familiar mouse on one side and a new mouse on the other. The GF mouse spent a comparable amount of time with the other two mice. This was in contrast with the behavior of microbiota-colonized mice, which spent more time with new mice than with familiar ones. The GF mouse seemed to spend more time in the empty room and next to objects compared to the other mice, and this is clearly atypical behavior in social animals. When the GF mouse was colonized, its behavioral patterns were partially normalized [63]. In a study by Adams et al., stools from 58 patients with ASD and 39 normal children were compared. In ASD patients, the levels of SCFAs, such as acetate, propionate, and valerate, were low. In addition, the level of bifidobacterial species was lower, while the level of *Lactobacillus* species was higher [64]. Kang et al. analyzed the bacterial 16S rDNA of 20 autistic children and found that levels of *Prevotella*, *Coprococcus*, and Veillonellaceae were significantly lower in samples from autistic children [65]. Kang et al. reported that a significant improvement persisted for 8 weeks after microbial transfer therapy when ASD symptoms were assessed in 18 pediatric ASD patients [66]. Hsiao et al. reported a significant improvement in communicative, stereotypic, anxiety-like, and sensorimotor behaviors when human commensal *Bacteroides fragilis* was administered to a maternal immune activation mouse model showing ASD symptoms, and *B. fragilis* was found to regulate various types of metabolites [67]. In a study by Golubeva et al., when prenatal stress was prolonged through repeated restraint stress in a dam within 14–20 embryonic days, male offspring showed a significant decrease in *Lactobacillus* spp. 4 months after birth, along with significant increases in those from the genera *Oscillibacter*, *Anaerotruncus*, and *Peptococcus* [68]. Table 2 summarizes the main characteristics of these trials.

### 5.4. Schizophrenia

Schizophrenia is a debilitating psychiatric disorder that causes physical and social morbidity. The disorder affects various aspects of daily life and typically persists for decades. The global burden of this disease is high [69].

Schwarz et al. compared differences in the fecal microbiota of 28 patients with first-episode psychosis (FEP) and 16 normal subjects. In the FEP group, there was a significant increase in *Lactobacillus* spp., which was highly correlated with the severity of each symptom. In addition, the greater the difference in microbiota, the worse the treatment response was after 12 months [70]. In a study by Castro-Nallar et al., differences in oropharyngeal microbiomes were compared between 16 SPR patients and 16 normal subjects. The SPR group exhibited a significant increase in microbial phyla including Proteobacteria, Firmicutes, Bacteroidetes, and Actinobacteria compared to the control group [71]. Table 3 summarizes the main characteristics of these trials.

### 5.5. Bipolar Disorder (BD)

BD is a multicomponent illness involving episodes of severe mood disturbance, neuropsychological deficits, immunological changes, and disturbances in functioning. It is one of the leading causes of disability worldwide and is associated with high rates of premature mortality from both suicide and medical comorbidities. To date, the pathogenesis of BD has not been fully elucidated. Interactions between genetic and environmental factors may play a role.

In a study by Coello et al. the differences in microbiota between 113 BD patients, 39 unaffected first-degree relatives, and 77 healthy controls were analyzed. There was a significant difference in the gut microbiota composition between BD patients and healthy controls (R^2^ = 1.0%, *p* = 0.008). In particular, *Flavonifractor* spp. were found in 61% of BD patients, 39% of unaffected relatives, and 39% of healthy controls. The odds ratio for the presence of *Flavonifractor* spp. in BD patients was 2.9 (95% CI: 1.6–5.2) [72]. A study by Painold et al. reported a negative correlation between microbial alpha diversity and illness duration of BD (R = −0.408, *p* = 0.007) through 16S RNA gene sequencing in 32 BD patients and 10 controls. The bacterial clades of BD patients were associated with inflammatory status, serum lipids, depressive symptoms, oxidative stress, and metabolic syndrome. In addition, Actinobacteria and Coriobacteria were significantly more abundant in BD patients than in healthy controls [73]. Aizawa et al. analyzed the fecal bacteria of 39 patients with BD and 58 healthy controls. Although there was no difference in bacterial count between the groups, a negative correlation was observed between *Lactobacillus* spp. strains count and sleep (R = −0.45, *p* = 0.01), as well as between *Bifidobacterium* spp. and cortisol levels (R = −0.39, *p* = 0.02) [74]. Evans et al. analyzed the stool microbiome of 115 BD patients and 64 healthy controls and found a significant difference in *Faecalibacterium* spp. between groups. In particular, there was a significant difference in the OUT fraction of *Faecalibacterium* spp. in terms of the self-reported health outcome using the Short Form Health Survey (SF12), Patient Health Questionnaire (PHQ9), Pittsburg Sleep Quality Index (PSQI), Generalized Anxiety Disorder Scale (GAD7), and Altman Mania Rating Scale (ASRM) [75]. Lu et al. used the concept of brain–gut coefficient of balance (B-G_CB_), which refers to the ratio of [oxygenated hemoglobin]./[ratio of *Bifidobacteria* to *Enterobacteriaceae* strains]., to compare the gut microbiota and brain function in patients with BD. The results of the study revealed high counts of *Faecalibacterium prausnitzii*, Bacteroides-Prevotella group, Atopobium Cluster, *Enterobacter* spp., and Clostridium IV in BD patients, but the Log_10_(B/E) value, which signifies microbial colonization resistance as the ratio of *Bifidobacteria* spp. and *Enterobacteriaceae* spp., was lower than that of the healthy controls. Based on these observations, it was suggested that the composition of the gut microbiota and its connectivity with brain function were altered in the BD group [76]. Table 3 summarizes the main characteristics of these trials.

### 5.6. Anxiety

Anxiety is commonly observed in mental disorders as well as in a variety of physical disorders, especially those related to stress. An increasing number of fundamental studies have indicated that the gut microbiota can regulate brain function through the gut–brain axis, and dysbiosis of the intestinal microbiota is related to anxiety.

In a study by Heijtz et al., GF mice exhibited an increase in motor activity and a decrease in anxiety-like behavior compared to SPF mice with normal gut microbiota in line with the notion that gut microbiota colonization may affect mammalian brain development [77]. Neufeld et al. performed elevated plus maze tests in GF mice in order to examine the anxiolytic effect. A decrease in N-methyl-D-aspartate receptor subunit NR2B mRNA expression was observed in the central amygdala, and an increase in the expression of BDNF, as well as a decrease in the expression of serotonin receptor 1A (5HT1A) in the hippocampus, were reported [78]. In a study by Davis et al., behavior and stress testing with regard to microbiota were performed in GF zebrafish larvae. The GF zebrafish demonstrated changes in locomotor- and anxiety-related behaviors, and the latter improved following treatment with the probiotic *Lactobacillus plantarum* [79]. In a study by Crumeyrolle-Arias et al., the behavioral response to social interaction in GF and SPF F344 male rats was assessed. Compared to SPF rats, GF rats exhibited significantly less sniffing time around an unknown partner in addition to a significant reduction in the frequency of visiting an aversive central area. Further, the serum corticosterone concentration of GF rats was 2.8 times higher under open-field stress compared to that of the SPF rats. CRF mRNA expression increased in the hypothalamus, and lower GR mRNA expression was observed in the hippocampus [80]. Bercik et al. administered drinking water containing nonabsorbable antimicrobials (neomycin, bacitracin, and pimaricin) for seven days to SPF BALB/c mice and observed a temporary change in microbiota composition, an increase in explorative behavior, as well as upregulated expression of hippocampal BDNF [81]. Hemmings et al. performed bacterial 16S ribosomal RNA sequencing in 18 PTSD patients and 12 trauma-exposed (TE) controls. There was no difference in alpha or beta diversity between the PTSD group and the TE group. Actinobacteria, Lentisphaerae, and Verruunomicrobia were the major phyla in the PTSD group. Further, the decrease in these taxa was correlated with the clinician-administered PTSD scale score (R = −0.387, *p* = 0.35) [82]. In a study by Messaoudi et al., a probiotic formulation (PF) containing a mix of *Lactobacillus helveticus* R0052 and *Bifidobacterium longum* R0175 was administered to healthy subjects before evaluation by the Hopkins symptom checklist (HSCL-90), hospital anxiety and depression scale (HADS), perceived stress scale, coping checklist (CCL), and 24 urinary free cortisol (UFC). The PF significantly decreased anxiety-like behavior, and a comparison with the control group revealed a significant difference in HSCL-90 (*p* < 0.5), HADS (*p* < 0.5), CCL (*p* < 0.5), and UFC (*p* < 0.5). Therefore, it was suggested that *L. helveticus* R0052 and *B. longum* R0175 have anxiolytic-like activity and beneficial psychological effects [83]. In a study by Rao et al., 39 patients with chronic fatigue syndrome were given either 24 billion colony-forming units of LcS or a placebo for 2 months, and the former led to a significant reduction in anxiety symptoms (*p* = 0.1) [84]. Table 4 summarizes the main characteristics of these trials.

### 5.7. Obsessive–Compulsive Disorder (OCD)

OCD is an anxiety disorder characterized by obsessive anxiety-producing thoughts often alleviated by compulsive behaviors, over which the patient has little control.

To identify the effects of probiotics on OCD, Kantak et al. injected BALB/cJ house mice with RU 24,969 followed by *L. rhamnosus* and saline to assess OCD-like behaviors (increased perseverative open-field locomotion, stereotypic turning, and marble burying). The group that received two weeks of probiotic administration exhibited a significant improvement in symptoms [85]. Shanahan et al. compared two weeks of *L. rhamnosus* and saline with four weeks of fluoxetine and found that the administration of *L. rhamnosus* was just as effective as fluoxetine administration in reducing OCD-like behaviors [86]. Savignac et al. conducted a stress-induced hyperthermia test, marble burying, elevated plus maze, open field, tail suspension test, and an FST after six weeks of daily *B. longum* 1714, *B. breve* 1205, and escitalopram administration to innately anxious BALB/c mice. Both the bifidobacterial and escitalopram groups exhibited a reduction in anxiety according to the marble burying test, while a reduction was only observed for the *B. longum* 1714 group based on stress-induced hyperthermia. In addition, *B. breve* 1205 reduced anxiety as assessed by the elevated plus maze, and *B. longum* 1714 exerted antidepressant-like effects based on the tail suspension test [87]. Turna et al. reported that OCD patients exhibited showed lower species richness/evenness and a relatively low abundance of butyrate-producing genera (*Oscillospira*, *Odoribacter*, and *Anaerostipes*) compared to healthy controls using 16S rRNA microbiota sequencing analysis performed on samples from 21 OCD patients and 21 healthy controls [88]. Table 5 summarizes the main characteristics of these trials.

### 5.8. Eating Disorder

Anorexia nervosa (AN) is a psychological and potentially life-threatening eating disorder. Patients typically suffer from an extremely low body weight relative to their height and body type.

Kleiman et al. conducted a Beck depression inventory, Beck Anxiety Inventory, Eating Disorder Examination Questionnaire, and 16S rRNA sequencing in 26 AN patients. The AN group exhibited a difference in taxa abundance and beta diversity, as well as significantly lower alpha diversity. In addition, their levels of depression, anxiety, eating disorders, and psychopathology were correlated with the composition and diversity of their intestinal microbiota [89]. Morita et al. performed 16S and 23S rRNA sequencing on 25 female AN patients (restrictive [*n* = 14]. and binge eating [*n* = 11].) and 21 control subjects. In AN patients, the levels of *Clostridium coccoides*, *Clostridium leptum*, *B. fragilis*, and *Streptococcus* spp. were significantly lower. In comparison to the control group, *B. fragilis* was significantly lower in both the anorexia nervosa restricting type (ANR) and anorexia nervosa binge eating type (ANBP), whereas *C. coccoides* was significantly lower only in the ANR group compared to the control group [90]. Mack et al. compared fecal microbiota by dividing AN patients into before recovery (*n* = 55), after weight gain (*n* = 44), and normal weight (*n* = 55) groups. In the AN patient group, the number of *Clostridium* clusters increased and butyrate-producing *Roseburia spp*. decreased. The concentration of branched-chain fatty acids, which is a known protein fermentation marker, was increased. After weight recovery, microbial richness increased, but the perturbation of intestinal microbiota and the SCFA profile could not be recovered [91]. Table 5 summarizes the main characteristics of these trials.

## 6. Conclusions

With the development of techniques for analyzing the human gut microflora, the importance of the microbiome for physical and mental health is increasingly acknowledged. The human brain and gut interact through the brain–gut axis. Psychosocial or psychological stress leads to cortisol secretion through the HPA gland axis, and this hormone changes intestinal permeability and the environment of gut microbes. In addition, the activation of the autonomic nervous system is associated with changes in gastrointestinal motility and the synthesis of various neurotransmitters. Conversely, dysbiosis of microorganisms in the gastrointestinal tract increases intestinal permeability, and various metabolites and inflammatory substances reach the brain through circulation, resulting in psychological symptoms such as depression, anxiety, cognitive decline, and a lack of social functioning. In particular, the activation of the immune response due to changes in the intestinal microflora causes the secretion of various inflammatory cytokines, which reach the brain, affecting psychological functions. Therefore, the brain–gut microbiome axis should be investigated as a bidirectional pathway, rather than emphasizing on the importance of either direction. In addition, the microbiome is known to produce or participate in the synthesis of various neurotransmitters and neuromodulators. However, most studies have been preclinical, and more human studies need to be conducted based on the results of animal studies. Increasing attention is being paid to treatment-resistant depression and treatment-resistant psychosis, for which it is difficult to achieve therapeutic benefits due to patients’ limited response to psychotherapy and psychiatric drugs. Further, the treatment of psychological disorders and the improvement of psychological symptoms by researching microbiomes may bring about innovative changes in clinical practice in the future.

## Figures and Tables

**Table 1 ijms-21-07122-t001:** Characteristics of prior studies investigating the relation between microbiome and Depression.

Psychiatric Disorder	Authors (Years)	Subject	Method	Result
Depression	Gareau et al. [50] (2017)	Pups (*n* = 12)	Separated from mother(MS) model 10 (8) probiotic organisms (two strains of *Lactobacillus* species)	Probiotic administration ameliorated the MS-induced gut functional abnormalities and reduced the elevated corticosterone levels
	Bharwani et al. [51] (2017)	Male C57BL/6 mice	*Lactobacillus rhamnosus* (JB-1)	Stress-induced anxiety behavior was reduced
				Lack of social interaction was prevented
	Burokas et al. [52] (2017)	Male C57BL/6J mice	Probiotics (fruco-oligosaccharides (Fos) and galacto-oligosaccharides(Gos))	Antidepressant and anxiolytic effect
				Reduced stress-induced corticosterone release
	Desbonnet et al. [53] (2010)	Rats	Maternal separation model	Immune response, behavioral deficit, and basal noradrenaline concentration were normalized
			*Bifidobacterium* or citalopram	
			Forced swimming test	
	Bravo et al. [46] (2011)	Male BALB/c mice (*n* = 36)	*L.rhamnosus rhamnosus* (JB-1)	Increased expression of GABA_B1b_ mRNA
				Reduction in stress-induced corticosterone and anxiety and depression-related behavior
	Ait-Belgnaoui et al. [54] (2012)	Female rats	Partial restraint stress (PRS) *L. farciminis* for 2 weeks	Induced hyperpermeability and endoxemia
				Prevention of neuroinflammation and stress response in the HPA axis
	Liang et al. [55] (2015)	Adult specific pathogen-free (SPF) rats	*Lactobacillus helveticus* N58 or citalopram	Decreased anxiety and depressive behaviors
			Behavioral test (Sucrose preference test, elevated-plus maze test, open-field test, object recognition test, object placement test)	Cognitive dysfunctions were improved at a level similar to citalopram group
	Bharwani et al. [56] (2016)	Male C57BL/6 mice	Chronic social defeat	Degree of social deficit was correlated with microbial community profile
			16S rRNA marker gene sequencing	Functional diversity was reduced in chronic social defeat group
	Kato-Kataoka et al. [57] (2016)	Double blind placebo-controlled trial (*n* = 24 vs. 23)	*Lactobacillus Casei Shirota* (LcS)	Improving physical symptoms in stressful situations
				Significant increase in fecal serotonin level

N: number.

**Table 2 ijms-21-07122-t002:** Characteristics of prior studies investigating the relationship between microbiome and Attention Deficit Hyperactivity Disorder, Autism Spectrum Disorder.

Psychiatric Disorder	Author (Years)	Subject	Method	Result
Attention deficit Hyperactivity Disorder (ADHD)	Howard et al. [59] (2011)	ADHD adolescents (*n* = 115)	Western dietary pattern vs. Healthy dietary pattern	Significantly higher number of diagnoses for ADHD in Western dietary pattern (OR = 2.21, 95% CI 1.18–4.13)
	Aarts et al. [60] (2017)	ADHD (*n* = 19)	16S rRNA marker gene sequencing	Difference in the relative abundance for the *bifidobacterium* genus
		HC (*n* = 77)		Increased in cyclohexadienyl dehydratase in ADHD
	Pärtty et al. [61] (2015)	Infants (*n* = 75)	*Lactobacillus rhamnosus GG* (ATCC 53103)	ADHD symptoms appeared in 6/35 (17.1%) in the placebo group but not in the probiotic group
			Compared gut microbiota after 3 weeks, 3,6,8,24 months and 13 years	
Autism Spectrum Disorder (ASD)	Borre et al. [63] (2014)	GF mice	Three separated room tests	GF mice seemed to spend more time with the empty room
				Behavioral patterns were partially normalized after colonized
	Adams et al. [64] (2011)	ASD children (*n* = 58)	Stool testing (bacterial and yeast culture test, lysozyme, lactoferrin, secretory IgA, elastase, digestion markers, short-chain fatty acid)	In ASD patients, levels of SCFA (acetate, proprionate, and valerate) and *bifidobacterial* species were low, *lactobacillus* species were high
		HC (*n* = 39)		
	Kang et al. [65] (2013)	ASD children (*n* = 20)	16S rDNA-targeting quantitative real-time PCR (qPCR)	Abundance of genera *Prevotella, Coprococcus, Veillonellaceae* was significantly reduced in ASD children
		Neurotypical children (*n* = 20)		
	Kang et al. [66] (2017)	ASD children (*n* = 18)	Fecal microbiota transplant (FMT)	Improvement of ASD symptoms persisted for 8 weeks after the treatment was completed
	Hsiao et al. [67] (2013)	Pregnant C57BL/6N mice	Maternal immune activation mouse model	Significant improved in communicative, stereotypic, anxiety-like and sensorimotor behaviors
			16S rRNA gene sequencing *Bacteroides fragilis*	
	Golubeva et al. [68] (2015)	Sprague–Dawley pregnant dams	Prenatal stress (repetitive restraint stress)	Significant decrease in *Latobacillus* 4 months after birth
			16S rRNA gene sequencing	

N: number, OR: Odds ratio, GF: Germ-free, HC: Healthy control.

**Table 3 ijms-21-07122-t003:** Characteristics of prior studies investigating the relationship between microbiome and Schizophrenia, Bipolar Disorder.

Psychiatric Disorder	Subject	Number	Method	Result
Schizophrenia	Schwarz et al. [70] (2018)	First episode psychosis (FEP) (*n* = 28)	16S rRNA gene sequencing	Numbers of *Lactobacillus* group bacteria were elevated in FEP-patients and significantly correlated with severity of symptom domains
		HC (*n* = 16)		
	Castro-Nallar [71] (2015)	Schizophrenia (*n* = 16)	Oropharynx microbiome	High-level differences were evident in *Proteobacteria, Firmicutes, Bacteroidetes and Antinobacteria*
		Healthy controls (*n* = 16)		
Bipolar Disorder (BD)	Coello et al. [72] (2019)	BD (*n* = 113)	16S rRNA gene sequencing	Significant differences of gut microbiota community among the group (R^2^ = 1.0%, *p* = 0.008)
		Unaffected first-degree relatives (*n* = 39)		*Flavonifractor* was found in 61% in BD, 39% in unaffected relatives and 39% in healthy control
		HC (*n* = 77)		Odd ratio of *flavonifractor* in BD was 2.9 (95% CI 1.6–5.2)
	Painold et al. [73] (2019)	BD (*n* = 32)	16S rRNA gene sequencing	Negative correlation between microbial alpha-diversity and illness duration of BD (R = 0.408, *p* = 0.07)
		HC (*n* = 10)		Bacterial clades associated with inflammatory status, serum lipid, depressive symptoms, oxidative stress, and metabolic syndromes
				*Actinobacteria* and *coriobacteria* were more abundant in BD
	Aizawa et al. [74] (2019)	BD (*n* = 39)	16S or 23S rRNA gene sequencing	*Lactobacillus* was associated with sleep (R = −0.45, *p* = 0.01)
		HC (*n* = 58)		*Bifidobacterium* was associated with cortisol level (R= −0.39, *p* = 0.02)
	Evans et al. [75] (2017)	BD (*n* = 115)	16S rRNA gene sequencing	Significant difference between groups in *feacalibacterium*
			Short-form Health Survey	
			Patient Health Questionnaire	
		HC (*n* = 64)	Pittsburg Sleep Quality Index	Significant differences between groups in *feacalibacterium* according to psychological scales
			Generalized Anxiety disorder	
			Altman Mania Rating Scale	
	Lu et al. [76] (2019)	BD (*n* = 36)	Brain–gut coefficient of	High count of *Feacalibacterium, prusnitzili, Bacteroides-prevotella, Atopobium, Enterobacter Spp, Clostridium IV* in BD
		HC (*n* = 27)	balance (B-G_CB_)	Composition of gut microbiome and connectivity with brain function are changed in BD

N: number, HC: Healthy control.

**Table 4 ijms-21-07122-t004:** Characteristics of prior studies investigating the relationship between microbiome and Anxiety Disorder.

Psychiatric Disorder	Authors (Years)	Subject	Method	Result
Anxiety Disorder	Heijtz et al. [77] (2011)	GF mice	Open field test	GF mice showed an increase in motor activity and a decrease in anxiety-like behavior compared with pathogen-free mice
			Elevated plus Maze test	
		SPF mice	Illumima Expression Array	
			Quantitative real-time PCR	
	Neufeld et al. [78] (2011)	GF mice	Elevated plus maze test	Decreased in NMDA receptor subunit NR2B mRNA expression
		SPF mice		Decrease expression of serotonin receptor 1A (5HT-1A)
	Davis et al. [79] (2016)	GF zebrafish	Behavioral and stress test	Anxiety-related behaviors improved following the *lactobacillus plantarum* administration
			16S rRNA sequencing	
	Crumeyrolle-Arias et al. [80] (2014)	GF rats	Open field stress test	GF rats showed significantly less sniffing time
		SPF pregnant F344 rats		Serum corticosterone concentration was 2.8 times higher in GF than in SPF
				CRF mRNA expression elevated and GR mRNA expression decreased in the hypothalamus
	Bercik et al. [81] (2011)	SPF BALB/C mice	Nonabsorbable antimicrobials	Temporary change in the composition of microbiota
			(neomycin, bacitracin, pimaricin)	Increase in explorative behavior and expression of hippocampal BDNF
	Hemmings et al. [82] (2017)	PTSD (*n* = 18)	16S rRNA sequencing	No difference in α or β diversity
		Trauma-exposed controls (TE) (*n* = 12)		*Actinobacteria, Lentisphaerae, Verrucomicrobia* were the major phyla in PTSD
				Decreased taxa were correlated with the clinician-administered PTSD scale score (r = −0.387, *p* = 0.35)
	Messaoudi et al. [83] (2011)	Probiotic formulation (PF) (*n* = 26)	Probiotic formulation (mixture of *lactobacillus helveticus R0052* and *Bifidobacterium longum R0175*)	PF group significantly decreased anxiety-like behavior
				Difference in Hopkins Symptom Check List (*p* < 0.05)
				Hospital Anxiety and Depression Scale (*p* < 0.05)
		Placebo (*n* = 29)		Perceived Stress Scale (*p* < 0.05)
				Copping Check List (*p* < 0.05)
				24 hr urinary free cortisol (*p* < 0.05)
	Rao et al. [84] (2009)	Chronic fatigue syndrome patients	24 billion colony-forming units of *lactobacillus casei* strain Shirota (Lcs)	Significant reduction in anxiety symptoms among taking the probiotics vs. controls (*p* = 0.01)
		-Treatment (*n* = 19)		
		-Placebo (*n* = 16)		

N: number, GF: Germ-free, SPF: Specific pathogen-free, PTSD: Post Traumatic Stress Disorder.

**Table 5 ijms-21-07122-t005:** Characteristics of prior studies investigating the relationship between microbiome and Obsessive Compulsive Disorder, Eating Disorder.

Psychiatric Disorder	Author (Years)	Subject	Method	Result
Obsessive–Compulsive Disorder (OCD)	Kantak et al. [85] (2014)	BALB/Cj mice	Ru 24969 and *L. rhamnosus*	Significant improvement in OCD-like symptoms (increased perseverative, open-field locomotion, stereotypic turning and marble-burying)
	Shanahan et al. [86] (2011)	Female Balb/cJ mice	*L. rhamnosus* Fluoxetine	Administration of *L. rhamnosus* was effective as fluoxetine administration in reducing OCD-like behaviors
	Savignac et al. [87] (2014)	Innately anxious BALB/C mice	-*Bifidobacterium longum 1714*	*Bifidobacterium longum 1714* group showed reduction in stress-induced hyperthermia and the tail suspension test
			-*Bifidobacterium breve 1205*	*B. breve 1205* reduced anxiety in the elevated plus-maze test
			-Escitalopram	
	Turna et al. [88] (2020)	OCD (*n* = 21)	16S rRNA sequencing	OCD group showed lower species richness/evenness and three butyrate-producing genera (*Oscillospira, Odoribacter, Anaerostipes*)
		HC (*n* = 21)		
Eating Disorder	Kleiman et al. [89] (2015)	AN (*n* = 26)	16S rRNA sequencing	AN group showed significantly low α diversity
			-Beck depression inventory	
			-Beck anxiety inventory	Psychopathology was correlated with the composition and diversity of microbiota
			-Eating disorder examination questionnaire	
	Morita et al. [90] (2015)	Female AN (*n* = 25)	16S and 23S rRNA sequencing	AN group showed *Clostridium coccoides, Clostridium leptum, Bacteroides fragilis, Streptococcus* were significantly lower
		restrictive (*n* = 14)		
		binge eating (*n* = 11)		
		HC (*n* = 21)		
	Mack et al. [91] (2016)	AN (*n* = 99)	16S rRNA sequencing	AN group showed the clostridium cluster increased and butyrate-producing *Roseburia* spp. decreased
		-before recovery (*n* = 55)		
		-after weight gain (*n* = 44)		After weight recovery, microbial richness increased, but perturbation of microbiota was not recovered
		HC (*n* = 55)		

N: number, HC: Healthy control, AN: Anorexia nervosa.

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
