# Peer review of "The Brain–Gut–Microbiome Axis in Psychiatry"

_ijms, 2020, doi:10.3390/ijms21197122_

Round 1
Reviewer 1 Report
The review clearly described the impact of brain-gut-microbiome axis on psychiatric disorders. I believe that the manuscript is well written and represents a well-done summary of the literature in the field. It is well structured and I enjoyed very much reading it. Therefore, I suggest the publication of the manuscript in the current version.
regarding the manuscript, I believe that the authors prepared a very good review strongly supported by the literature.
In case, if possible, a Figure would help readers in summarizing the most important topics of the manuscript.
Author Response
Dear reviewer,
Wereally appreciate reviewers’ comments before. We think the comments have made our article more robust than before.
According to the comments, we rivesd the manuscripts. (The revised contents are marked in red)
We revised the overall English grammar and supplemented the readability, and the Table 1-5 has been added.
Thank you again, for your comment.
Reviewer 2 Report
Authors in presented paper decided to investigate relationship between gut microbiota and function of brain-gut axis in physiology and pathophysiology of psychiatric deseases.
Broad comments
The subject of presented paper is interesting and current, gut microbiota is a hot research topic over the last few years. Authors at first presented overall description of brain-gut axis, role of microbiota in neurotransmitters action, and then reviewed role of gut microbiota in several diseases, what is as asset of this review. Despite that, presented manuscript requires moderate English editing, with substantive explanation of a few assumptions and what is the most imporant paragraph 5 needs to be revised, because it is not clear in the present form and not readable.
Specific comments
- English edtiting in needed including:
line 103 patients with depression,
lines 123-124 the sentence can be rewritten to .. substances necessary for neurotransmitter synthesis.
line 166 - behaviour ... are functions not diseases so it should be "CNS functions"
line 177 - in association of what of vagus nerve? its? activity? it should be explained
line 286 - MDD was used a few times, but was not explained what it stands for, it should be clear what kind of abbreviation it is.
line 339 - administration is missing after fluoxetine
- line 46-48: In this sentence authors state, that human brain contains 1kg of bacteria. To my knowledge brain and CNS in standard physiological state is free from bacteria and aseptic. Did authors mean gut? I can agree that it is said, that bacteria in the gut can weight up to 1kg.
- lines 125-126: The order of this sentence is misleading. It seems like authors state that tryptophan is synthetized by gut-bacteria and then they metabolize it to form indoles, serotonin and melatonin. Later authors write, that excesive bacteria metabolism decreases tryptophan availability for host. These 2 assumptions are not consistent. To my knowledge gut bacteria use tryptophan administered in host's diet. Tryptophan is an exogenous amino acid for humans. If there was a de novo synthesis of tryptophan in bacteria there might be only minimal changes in endogenous metabolism of tryptophan by the host. This part of the manuscrpit should be rewritten with apropriate references concerninig metabolism of tryptophan and possible changes in its metabolism caused by high-tryptophan and tryptophan-free diet.
- Lines: 278-285: in this paragraph Bipolar disorder should be reviewed, and in those lines authors cited results concerning healthy subjects. Why is that?
- Paragraph 5: This part of the manuscript might be the most important for viewers, but is written to technically, almost every subunit starts with "this author did this and that, that author did ...". It is not easy to read and not that interesting in presented form. Each disease should have a 1-2 sentence introduction concering what is specific for this disease, as it is written in 5.3 ADS. Additionally, each subunit should have at least 1 sentence at the end as a summary or proposed practical implementation of reviewed aspects.
- I believe, that article could be enriched by adding at least one figure or table as a summary of reviewed diseases with additional information of main changes in gut microbiota composition in every disease.
Author Response
Dear reviewer,
Wereally appreciate reviewers’ comments before. We think the comments have made our article more robust than before.
According to the comments, we rivesd the manuscripts. (The revised contents are marked in red)
We revised the overall English grammar and supplemented the readability, and the Table 1-5 has been added.
- Presented manuscript requires moderate English editing, with substantive explanation of a few assumptions and what is the most imporant paragraph 5 needs to be revised, because it is not clear in the present form and not readable.
According to the reviewer’s comments, We revised the overall English grammar and supplemented the readability
- line 103 patients with depression
According to the reviewer’s comments, the manuscrips have been modified. (Line 99)
- lines 123-124 the sentence can be rewritten to .. substances necessary for neurotransmitter synthesis.
According to the reviewer’s comments, the manuscrips have been modified. (Line 119-120)
- line 166 - behaviour ... are functions not diseases so it should be "CNS functions"
According to the reviewer’s comments, the manuscrips have been modified. (Line 161-162)
- line 177 - in association of what of vagus nerve? its? activity? it should be explained
According to the reviewer’s comments, the manuscrips have been modified. (Line 174)
- line 286 - MDD was used a few times, but was not explained what it stands for, it should be clear what kind of abbreviation it is.
The term MDD has been deleted.
- line 339 - administration is missing after fluoxetine
According to the reviewer’s comments, the manuscrips have been modified. (Line 361)
- line 46-48: In this sentence authors state, that human brain contains 1kg of bacteria. To my knowledge brain and CNS in standard physiological state is free from bacteria and aseptic. Did authors mean gut? I can agree that it is said, that bacteria in the gut can weight up to 1kg.
According to the reviewer’s comments, the manuscrips have been modified. (Line 44)
- lines 125-126: The order of this sentence is misleading. It seems like authors state that tryptophan is synthetized by gut-bacteria and then they metabolize it to form indoles, serotonin and melatonin. Later authors write, that excesive bacteria metabolism decreases tryptophan availability for host. These 2 assumptions are not consistent.
According to the reviewer’s comments, the manuscrips have been modified. (Line 121-123)
- Lines: 278-285: in this paragraph Bipolar disorder should be reviewed, and in those lines authors cited results concerning healthy subjects. Why is that?
According to the reviewer’s comments, the manuscrips have been modified. (Line 282-313)
- Paragraph 5: This part of the manuscript might be the most important for viewers, but is written to technically, almost every subunit starts with "this author did this and that, that author did ...". It is not easy to read and not that interesting in presented form.
According to the reviewer’s comments, We revised the overall English grammar and supplemented the readability
- Each disease should have a 1-2 sentence introduction concering what is specific for this disease, as it is written in 5.3 ADS. Additionally, each subunit should have at least 1 sentence at the end as a summary or proposed practical implementation of reviewed aspects.
According to the reviewer’s comments, we added contents. (The revised contents are marked in red)
- I believe, that article could be enriched by adding at least one figure or table as a summary of reviewed diseases with additional information of main changes in gut microbiota composition in every disease.
According to the reviewer’s comments, we added Table 1-5.
Reviewer 3 Report
This is an interesting review dealing with the crucial interactions between the brain and the gut microbiota participating in the development of psychiatric disorders. However, the presentation of cited evidence is somewhat confusing, and the readability of the discussion should be improved.
- The sentence “The adult human brain contains approximately 1 kg of bacteria, which is similar to the weight of the brain itself” is entirely wrong, “The human gut harbors a dynamic and complex microbial ecosystem, consisting of approximately 1 kg of bacteria in the average adult, approximately the weight of the human brain,” -according to cited work by Dinnan et al. (2015).
- Lines 122-123: please confirm the information that “Some essential vitamins, such as vitamins K, B2, B9, and B12, synthesized by microbiota have a neuroprotective effect on the CNS” and provide an adequate reference.
- Line 125: Microbiota is not as “a precursor of the synthesis of indole, serotonin, and melatonin.” It produces these bioactive products, including several indole-containing metabolites and 5-HT.
- The description of all cited studies should be completed with the subjected species. Data from studies in humans should be clearly highlighted. Authors could consider tabulating the demonstrated evidence with a clear division into research models and outputs.
- Data on humans should also support the effects of probiotics on the obsessive compulsive disorder.
Author Response
Dear reviewer,
Wereally appreciate reviewers’ comments before. We think the comments have made our article more robust than before.
According to the comments, we rivesd the manuscripts. (The revised contents are marked in red)
We revised the overall English grammar and supplemented the readability, and the Table 1-5 has been added.
- The sentence “The adult human brain contains approximately 1 kg of bacteria, which is similar to the weight of the brain itself” is entirely wrong, “The human gut harbors a dynamic and complex microbial ecosystem, consisting of approximately 1 kg of bacteria in the average adult, approximately the weight of the human brain,” -according to cited work by Dinnan et al. (2015).
According to the reviewer’s comments, the manuscrips have been modified. (Line 44)
- Lines 122-123: please confirm the information that “Some essential vitamins, such as vitamins K, B2, B9, and B12, synthesized by microbiota have a neuroprotective effect on the CNS” and provide an adequate reference.
According to the reviewer’s comments, the manuscrips have been modified. (Line 118-119 and Line 477; ref. 28)
- Line 125: Microbiota is not as “a precursor of the synthesis of indole, serotonin, and melatonin.” It produces these bioactive products, including several indole-containing metabolites and 5-HT.
According to the reviewer’s comments, the manuscrips have been modified. (Line 121-123)
- The description of all cited studies should be completed with the subjected species. Data from studies in humans should be clearly highlighted. Authors could consider tabulating the demonstrated evidence with a clear division into research models and outputs.
According to the reviewer’s comments, we added Table 1-5.
- Data on humans should also support the effects of probiotics on the obsessive compulsive disorder.
According to the reviewer’s comments, we added data. (Line 369-372)
Round 2
Reviewer 2 Report
Authors revised manuscript entitled: "Brain-Gut Microbiome axis in Psychiatry" according to reviewer's suggestions and submitted it in a corrected form.
English grammar and style correction has been implemented, what improved readability of this article.
There are still small spelling errors that need to be corrected:
In Table 1; Table 3 and Table 5 results sections should be spell checked and corrected
Small substantive errors have been corrected, including section concerning tryptophan metabolism.
Section about bipolar disease has been rewritten and now is more clear for the readers.
Authors included small introduction and summary for each subheading in part 5th of the manuscript, what made the whole article more interesting and improved its structure.
By adding 5 tables authors made it easier for readers to find what is interesting about each reviewed disease.
Overall I am pleased with the revised version of the article. I believe , that when corrected it will be a good contribution to research of associations between gut microbiota and human's wellbeing.
Author Response
Dear Reviewer 2
I really appreciate reviewer’s comments before. I think the comments have made our article more robust than before.
In Table 1; Table 3 and Table 5 results sections should be spell checked and corrected
- According to the reviewer’s comments, the tables have been corrected. (Table 1-5; marked in red)
Reviewer 3 Report
The manuscript has been significantly improved. However, it still needs to be checked the consistency of the cited authors' names in the text, tables, and references list. In particular, it is necessary to refer to items 53, 77, and 81.
Author Response
Dear Reviewer 3
I really appreciate reviewer’s comments before. I think the comments have made our article more robust than before.
The manuscript has been significantly improved. However, it still needs to be checked the consistency of the cited authors' names in the text, tables, and references list. In particular, it is necessary to refer to items 53, 77, and 81.
-> According to the reviewer’s comments, the manuscrips have been corrected. (Line 223, 241, 335, 344, 362, 368, 643, and Table 1-5; marked in red)